# A nationwide school fruit and vegetable policy and childhood and adolescent overweight: A quasi-natural experimental study

Bente Øvrebø[1,2,3]*, Tonje H. Stea[4,5], Ingunn H. Bergh[2], Elling Bere[1,2,3], Pål Surén[6], Per Magnus[7], Petur B. Juliusson[8,9,10], Andrew K. Wills[11,12]

1 Department of Sport Science and Physical Education, University of Agder, Kristiansand, Norway, 2 Department of Health and Inequalities, Norwegian Institute of Public Health, Oslo, Norway, 3 Centre for Evaluation of Public Health Measures, Norwegian Institute of Public Health, Oslo, Norway, 4 Department of Health and Nursing Sciences, University of Agder, Kristiansand, Norway, 5 Department of Child and Adolescence Mental Health, Sørlandet Hospital, Kristiansand, Norway, 6 Department of Child Health and Development, Norwegian Institute of Public Health, Oslo, Norway, 7 Centre for Fertility and Health, Norwegian Institute of Public Health, Oslo, Norway, 8 Department of Health Registry Research and Development, Norwegian Institute of Public Health, Bergen, Norway, 9 Department of Clinical Science, University of Bergen, Bergen, Norway, 10 Children and Youth Clinic, Haukeland University Hospital, Bergen, Norway, 11 Faculty of Health Sciences, University of Bristol, Bristol, United Kingdom, 12 Department of Nutrition and Public Health, University of Agder, Kristiansand, Norway

* bente.ovrebo@uia.no

## Abstract

### Background

School free fruit and vegetable (FFV) policies are used to promote healthy dietary habits and tackle obesity; however, our understanding of their effects on weight outcomes is limited. We assess the effect of a nationwide FFV policy on childhood and adolescent weight status and explore heterogeneity by sex and socioeconomic position.

### Methods and findings

This study used a quasi-natural experimental design. Between 2007 and 2014, Norwegian combined schools (grades 1–10, age 6 to 16 years) were obligated to provide FFVs while elementary schools (grades 1–7) were not. We used 4 nationwide studies ($n$ = 11,215 children) from the Norwegian Growth Cohort with longitudinal or cross-sectional anthropometric data up to age 8.5 and 13 years to capture variation in FFV exposure. Outcomes were body mass index standard deviation score ($BMI_{SDS}$), overweight and obesity (OW/OB), waist circumference (WC), and weight to height ratio (WtHR) at age 8.5 years, and $BMI_{SDS}$ and OW/OB at age 13 years. Analyses included longitudinal models of the pre- and post-exposure trajectories to estimate the policy effect. The participation rate in each cohort was >80%, and in most analyses <4% were excluded due to missing data. Estimates were adjusted for region, population density, and parental education. In pooled models additionally adjusted for pre-exposure $BMI_{SDS}$, there was little evidence of any benefit or unintended consequence from 1–2.5 years of exposure to the FFV policy on $BMI_{SDS}$, OW/OB, WC, or WtHR in either sex. For example, boys exposed to the FFV policy had a 0.05 higher $BMI_{SDS}$ (95%

**Citation:** Øvrebø B, Stea TH, Bergh IH, Bere E, Surén P, Magnus P, et al. (2022) A nationwide school fruit and vegetable policy and childhood and adolescent overweight: A quasi-natural experimental study. PLoS Med 19(1): e1003881. https://doi.org/10.1371/journal.pmed.1003881

**Data Availability Statement:** Data cannot be shared publicly because of General Data Protection Regulation (GDPR) as it contains personal data which is potentially identifying participant data. The

Personal Data Regulations and Health Research Act (§ 7) in Norway and GDPR in the EU restrict sharing of these data. The Norwegian Institute of Public Health administer the main data used in this study. External researchers can apply for access to indirectly identifiable data based on appropriate legal bases for processing the data in accordance with GDPR Article 6(1) and 9(2). To apply for access to the data: https://www.fhi.no/en/more/access-to-data/applying-for-access-to-data/. For more information about access to the data used in this study or questions regarding data requests: https://www.fhi.no/div/helseundersokelser/vekstkohorten/tilgang-til-data-fra-vekstkohorten/ (website in Norwegian) or contact vekstkohorten@fhi.no.

**Funding:** This work was supported by the Norwegian Research Council (grant number 260408/H10). Authors AKW at the University of Bristol, United Kingdom; EB at the University of Agder, Norway; and PM at the Norwegian Institute of Public Health received funding from the Norwegian Research Council. The Norwegian Research Council had no role in the design, analysis or writing of this article.

**Competing interests:** The authors have declared that no competing interests exist.

**Abbreviations:** BMI, body mass index; BMI$_{SDS}$, body mass index standard deviation score; DAG, directed acyclic graph; FFV, free fruit and vegetable; FV, fruit and vegetable; MLM, multilevel model; NCGS, Norwegian Childhood Growth Study; NFFV, no free fruit and vegetable; NYGS, Norwegian Youth Growth Study; OR, odds ratio; OW/OB, overweight and obesity; RCT, randomized controlled trial; WC, waist circumference; WtHR, waist to height ratio.

CI: −0.04, 0.14), a 1.20-fold higher odds of OW/OB (95% CI: 0.86, 1.66) and a 0.3 cm bigger WC (95% CI: −0.3, 0.8); while exposed girls had a 0.04 higher BMI$_{SDS}$ (95% CI: −0.04, 0.13), a 1.03 fold higher odds of OW/OB (95% CI: 0.75, 1.39), and a 0-cm difference in WC (95% CI: −0.6, 0.6). There was evidence of heterogeneity in the policy effect estimates at 8.5 years across cohorts and socioeconomic position; however, these results were inconsistent with other comparisons. Analysis at age 13 years, after 4 years of policy exposure, also showed little evidence of an effect on BMI$_{SDS}$ or OW/OB. The main limitations of this study are the potential for residual confounding and exposure misclassification, despite efforts to minimize their impact on conclusions.

## Conclusions

In this study we observed little evidence that the Norwegian nationwide FFV policy had any notable beneficial effect or unintended consequence on weight status among Norwegian children and adolescents.

---

## Author summary

### Why was this study done?

- To promote a healthy diet, from 2007 to 2014 a nationwide free fruit and vegetable policy ensured that a daily piece of free fruit or vegetable was available to all children in Norwegian combined schools (covering grades 1–10, age 6 to 16 years).

- Studies on the potential benefits or consequences of such fruit and vegetable policies are important in improving public health efforts to tackle childhood overweight and obesity.

### What did the researchers do and find?

- The policy rollout resulted in provision of free fruit and vegetables to children in combined elementary and secondary schools, while children in pure elementary schools were not exposed to the free fruit and vegetable policy.

- We exploited this quasi-natural experimental design to assess whether there was any evidence of an effect of up to 4 years of exposure to the free fruit and vegetable policy on weight-related outcomes.

- Using data from 11,215 Norwegian children and early adolescents, we observed little evidence of any beneficial or unintended impact from exposure to the free fruit and vegetable policy on weight outcomes in either boys or girls at age 8.5 and 13 years.

**What do these findings mean?**

- Our findings suggest that a national free fruit and vegetable policy alone is unlikely to have a notable impact on population childhood weight outcomes; however, such policies may promote a healthy diet without unintended consequences.

## Introduction

Schools are an optimal setting for health promotion due to the potential to reach all children regardless of socio-demographics [1]. The World Health Organization has highlighted the importance of school nutrition policies in promoting a healthy diet, and the European Union has implemented a school fruit and vegetable (FV) policy to enhance adherence to nutritional recommendations and prevent overweight and obesity (OW/OB) [2–4]. In 2020–2021, 26 of 44 European countries distributed FVs to schoolchildren [5]. Similar programs have been implemented elsewhere [6–8].

National school FV programs have been shown to increase FV consumption among children [6,7,9], but our understanding of their effect on childhood obesity outcomes is limited [8,10]. Meta-analyses and systematic reviews of randomized controlled trials (RCTs) indicate that increased FV consumption may promote weight loss and prevent weight gain [11,12], as the FVs consumed may substitute for more energy-dense foods [13,14]. However, school food provision, such as school lunch programs, could increase weight [15]. Given the public health challenge of childhood OW/OB [16–18], information about the possible benefits or unintended consequences of school dietary interventions is clearly important. Despite this, there are very few evaluations of school FFV provision. Two studies, with 7- and 14-year follow-up, comparing self-reported weight status of Norwegians who had received 1 elementary school year of free FVs (FFVs) compared to controls found little evidence for an effect on overweight although the sample size in both studies was small [10,19]. Another study investigated the effect of a FFV program in low-income public schools in Arkansas, US [8]. This study, set in a population with a high prevalence of childhood obesity, showed a reduction in body mass index (BMI) and obesity. Larger, more population-wide evaluations of school FFV provision on OW/OB are clearly needed [10,19].

From 2007 to 2014, the Norwegian government implemented a nationwide school FFV provision policy for lower secondary schools (pupils age 13–15 years). Since approximately one-third of elementary schools are combined with lower secondary schools, elementary age children (6–12 years) attending a combined school also received FFVs while those attending a pure elementary school did not receive FFVs, providing a nationwide quasi-natural experimental setting for policy evaluation [20]. Our objective was to assess whether exposure to the nationwide FFV policy for up to 4 years from starting school resulted in any benefits or unintended consequences with respect to childhood and early adolescent BMI and weight status. We also assessed if the response differed by sex and socioeconomic position.

## Methods

### The FFV policy and analytical design

From August 2007 to June 2014, all combined schools (grades 1–10) in Norway were obligated by the FFV policy to provide pupils with a daily portion of FVs while all pure elementary

schools (grades 1–7) were not (referred to as no FFV [NFFV] schools). The FFV policy was not accompanied by other components beyond FV provision. The portion typically consisted of an apple, pear, banana, orange, clementine, kiwi, carrot, or nectarine and was usually provided during lunch. The study design was driven by the policy rollout and the availability of datasets from the Norwegian Growth Cohort. The analysis strategy was planned a priori, but we did not register a protocol due to a combination of delays in data access and fallout from the COVID-19 pandemic. Any secondary or post hoc analyses that were done in response to the results or the review process are defined in the text. This study is reported as per the Strengthening the Reporting of Observational Studies in Epidemiology (STROBE) guideline (S1 Checklist).

Four nationwide cohorts that are part of the Norwegian Childhood Growth Study (NCGS) and Norwegian Youth Growth Study (NYGS) were used to capture variation in FFV policy exposure. The NCGS is a repeated cross-sectional survey of height, weight, and waist circumference (WC) of 8-year-old children (grade 3) conducted in schools in 2010, 2012, and 2015. The NYGS is similar but was conducted in 2017 on 13-year-olds (grade 8) and only for height and weight. We refer to these as the 2010, 2012, 2015, and 2017 cohorts. We also obtained repeated height and weight measurements recorded during the routine national health examinations scheduled from birth to 6 years of age for the 2010 and 2015 cohorts and from birth to 8 years of age for the 2017 cohort (S1 Fig shows a schematic of the study design). These cohorts allow several comparisons to assess the consistency of the evidence and strengthen causal inference. First, within each cohort there is variation in whether a child attended a FFV school or a NFFV school. Second, there is variation in the duration of exposure between some cohorts. Third, 2 of the cohorts were exposed for the same duration of exposure (2010 and 2012 cohorts), providing replication. Fourth, longitudinal information from 3 of the cohorts allow comparisons of the outcome trajectories before the intervention.

## Participants

Both the NCGS and NYGS used a 2-stage sampling scheme to obtain a nationally representative sample. In the first stage, 10 out of 19 counties were sampled from the geographical regions in Norway. In the second stage, schools were randomly sampled within each county. In the NCGS, the same 130 schools were invited to participate in 2010, 2012, and 2015, and between 123 to 126 schools agreed; in the NYGS, 150 out of 159 secondary schools participated. All third graders in participating schools were sampled in the NCGS cohorts, while 1 grade 8 class per school was sampled in the NYGS. The individual-level participation rate was >80% in the NCGS cohorts (2010, $n = 3,182$; 2012, $n = 3,508$; 2015, $n = 3,338$). The individual participation rate in the NYGS 2017 is unknown ($n = 1,907$). Additional information about the NCGS and NYGS can be found elsewhere [21,22].

## Data collection

**Anthropometry.** Height (to the nearest 0.1 cm), weight (to the nearest 0.1 kg), and WC (to the nearest 0.1 cm) were measured by school nurses during the fall for all cohorts using similar protocols (WC was not assessed in the 2017 cohort). The routine anthropometrics from health records were measured by nurses in health centers and the School Health Service. In Norway these measurements are scheduled at birth and 6 weeks; 3, 6, 9, 12, 15, 18, and 24 months; and 3, 4, 6, 8 (grade 3), and 13 years (grade 8). There is fluctuation around these target ages, and some appointments are missed (see S2 Fig). All height and weight values were cleaned using a longitudinal algorithm that checked for logical errors and internally

inconsistent values [23]. Full details of these quality assurance processes are described elsewhere [21,22].

**School information.**    School names, extracted from questionnaires completed by school nurses, were linked with the national school registry to determine whether schools were combined (FFV) or pure elementary (NFFV) schools. This questionnaire was received from all schools in the NCGS, and 137/150 schools in the NYGS. Information on elementary school affiliation for the grade 8 participants in the NYGS was obtained by parents as part of the consent form.

**Other data.**    National personal identification numbers were used to link children with records from the Medical Birth Registry of Norway and Statistics Norway. Parental education was used as an indicator for socioeconomic position. We used the highest parental education (mother or father) when the child was 4 years old, i.e., prior to policy exposure. Education was collapsed into 2 levels: higher education (education in university/college) or high school or less. Other classifications did not alter the main results at all (details in S6 Text). Information on county and health region (Northern, Central, Western, and Southern/Eastern) were used as markers of geographical location. A 3-category population density marker of school placement was obtained: urban (municipalities with a population > 50,000), semi-urban (municipalities with a population between 15,000 and 50,000), and rural (municipalities with a population < 15,000).

## Outcomes

Outcomes were BMI and overweight including obesity (OW/OB) in the third (age approximately 8.5 years) and eighth grade (age approximately 13 years), and WC and waist to height ratio (WtHR) in the third grade. To meet the linearity assumption of the main analytical models, an internally standardized age- and sex-adjusted BMI standard deviation score ($BMI_{SDS}$) was created [24]; modeling on the raw ($kg/m^2$) or externally standardized scale did not meet this assumption (see S1 Text for more details). Age- and sex-specific OW/OB was classified using the International Obesity Task Force cutoffs for BMI [25].

## Exposure classification

For the 2010, 2012, and 2015 cohorts, children attending a combined school at recruitment (third grade) were classified as exposed to the FFV policy. For the 2017 cohort (recruited in grade 8), children were classified as exposed if they attended a combined school during primary years. This classification does not account for children who were exposed to both school types due to moving schools; however, based on information in the 2017 cohort, we estimate that this occurs in less than 4% of children (see S2 Text). For the outcomes in third grade, this corresponds to 2–2.5 school years of exposure in the 2010, 2012, and 2017 cohorts and 1 year of exposure in the 2015 cohort. For the outcomes in grade 8 in the 2017 cohort, this corresponds to 4 school years of exposure. As the first day of school for Norwegian first graders is in August of the year children turn 6, the earliest age at which any child would have received school FFVs is 5 years and 7 months.

## Estimating the FFV policy effect

For $BMI_{SDS}$ and OW/OB, where longitudinal data were available (cohorts 2010, 2015, and 2017), 2 approaches were used to estimate the FFV policy effect. The first, illustrated in Fig A in S3 Text, is similar to a comparative interrupted time series analysis [26]. The pre- and post-intervention slopes in each group were modeled with linear splines and a knot at the pre-exposure age 5.5 years. The counterfactual is the trajectory that the FFV group would have taken in

the absence of the intervention and is estimated by the change in slopes in the NFFV group. The between-group difference in the pre–post difference in slopes is thus an estimate of the FFV policy effect. This can be parameterized as:

$$E(Y) = \beta_0 + \beta_1 S_1 + \beta_2 S_2 + \gamma_0 I + \gamma_1 I * S_1 + \gamma_2 I * S_2 \tag{1}$$

where $I$ is a binary variable indicating FFV exposure, and $S_1$ and $S_2$ are linear splines of age centered at the pre-intervention knot (additional details in S3 Text). $\beta_0$, $\beta_1$, and $\beta_2$ describe the outcome, $E(Y)$, at 5.5 years and the pre- and post-intervention slopes, respectively, in the control group. $\gamma_0$, $\gamma_1$ and $\gamma_2$ are the mean difference in intercept at 5.5 years and mean difference in pre- and post-intervention slopes, respectively, between the FFV and NFFV groups. Where pre-intervention slopes were similar, $\gamma_1$ was removed and $\gamma_2$ is the estimate of the policy effect. Where the pre-intervention slopes were different (as estimated by $\gamma_1$), $\gamma_2 - \gamma_1$ is the effect estimate, but in this situation, where pre-intervention slopes are not parallel, the counterfactual that slopes would have changed in the same way as the controls is less credible. Similar reasoning applies when there is a large difference in the pre-intervention intercept ($\gamma_0$). Hence a second approach that adjusts for the pre-intervention value of the outcome was also estimated:

$$E(Y) = \beta_0 + \beta_1 Y_{\text{PRE}} + \delta_1 I \tag{2}$$

Here, $Y_{\text{PRE}}$ is the closest available measurement before the introduction of the FFV exposure (5.5 years), and $\delta_1$ is an estimate of the FFV effect (the difference in $Y$ between groups after accounting for baseline differences). To estimate the effect at 13 years in 2017, Eqs 1 and 2 were extended in a separate model to include an extra knot at age 8.5 years (see S3 Text). For the WC and WtHR outcomes, where only a single measure of the outcome was available, the FFV policy effect estimator simplifies to a post-intervention between-group comparison (i.e., Eq 2 without $\beta_1$). Other potential confounders were added to these models (explained below).

## Analytical dataset

The pre-intervention slopes were modeled from age 2 years. To remove measurement clumping and minimize selection bias, if an individual had more than 1 measure at a target age, the value closest to the median age at each target assessment was selected. To ensure that the pre- and post-exposure slopes were demarcated by unexposed and exposed data points and avoid bias in estimating the 2 slopes, measures from age 5.7 years to 7 years were not included (see S3 Text for more details). More than 69% of individuals included in the analysis contributed at least 3 repeated measures.

## FFV policy allocation and estimating a causal effect

Allocation of the FFV policy could not be considered "as if" random. Combined (FFV) schools are more likely to be in areas of lower population density compared to pure elementary (NFFV) schools and are thus more common in rural regions of Norway such as the Northern region (see S4 Text). A directed acyclic graph (DAG) was thus used to inform which variables to adjust for to obtain a causal estimate of the policy effect (S5 Text; Fig A in S5 Text). Based on the DAG and testing the assumptions it encodes, the following variables were deemed sufficient to adjust for: region, population density, cohort, and parental education. The DAG also suggests parental education and sex may modify the effect of the FFV policy since they may affect whether or not the FVs are consumed and/or any induced dietary change. We also consider a separate and additional adjustment for pre-intervention BMI as this is a marker of the obesogenic environment of the child.

## Analyses

**FFV allocation and pre-intervention comparisons.** Characteristics prior to exposure (sex, parental education, region, and population density) were described by cohort and by FFV allocation. The pre-intervention slopes and intercepts of the $BMI_{SDS}$ and OW/OB outcomes were compared between groups using multilevel models (MLMs), and the marginal unadjusted and adjusted (described below) trajectories were plotted.

**Main analysis.** Analyses were stratified by cohort (due to differences in exposure duration), and sex (see DAG; Fig A in S5 Text), and pooled estimates were also produced. To make use of all available outcome data and account for the hierarchical structure, MLMs were used with random intercepts for each school and child, and random slopes for each child for the $BMI_{SDS}$ outcome. Autocorrelation in the $BMI_{SDS}$ models was handled using a first order autoregressive structure. A logit MLM with maximum likelihood and adaptive Gauss–Hermite quadrature estimation was used for the OW/OB outcome.

For the longitudinal cohorts (2010, 2015, and 2017), 3 sets of models were estimated: (1) an unadjusted model (crude); (2) a model adjusting for region, population density, and parental education (adjusted); and (3) a model with additional adjustment for pre-intervention $BMI_{SDS}$ (+pre-intervention adjusted). Potential confounders were allowed to affect intercepts and slopes, and pooled models included similar terms for cohort. For the cross-sectional WC and WtHR outcomes, only the crude and adjusted models could be estimated using the 2010, 2012, and 2015 cohorts. To assess potential effect modification by socioeconomic position, similar models were estimated but stratified by parental education (higher education or high school or less), with Wald tests of the interaction terms.

Effect estimates are reported comparing the difference in outcome at age 8.5 years and age 13 years between FFV exposure and the counterfactual (as estimated using NFFV schools). As WC was not measured in the NYGS, WC and WtHR outcome estimates could not be estimated at age 13 years. All results are displayed in forest-style plots to visualize heterogeneity.

**Supplemental and sensitivity analyses.** The Norwegian Directorate of Health and the Norwegian Fruit and Vegetable Marketing Board offer a national school FV subscription program that provides schools with the opportunity to offer FVs with parental payment. As all pure elementary schools (NFFV schools) were free to decide whether to offer parental paid FVs, we conducted a sensitivity analysis where we excluded children from the combined (NFFV) schools (151/335 schools; 2,022/6,168 children) that had offered the paid subscription program during at least 1 of the first 3 years of school, as ascertained from the Norwegian Fruit and Vegetable Marketing Board. If the FFV policy had a causal effect, estimates from this analysis would be expected to be stronger than those from the main analysis. All post hoc analyses were done as sensitivity analyses to check the robustness of any findings. These, alongside any analyses done in response to the review process, are defined as such in the text. Other sensitivity analyses were also performed to assess the robustness of findings to the analytical strategy; these are outlined in S5 Table.

## Ethics

Data are from the Norwegian Growth Cohort. This consists of the NCGS and NYGS, both conducted by the Norwegian Institute of Public Health in collaboration with the School Health Service and in accordance with the Helsinki Declaration. Ethical approval and research clearance were obtained from the Regional Committee of Medical and Health Research Ethics (2017/431 and 2010/938), and the research was approved by the Norwegian Data Inspectorate. Detailed information about the studies (NCGS and NYGS) was sent to parents or guardians

prior to each survey, and the School Health Service obtained written informed consent from parents or other legal guardians on behalf of the Norwegian Institute of Public Health.

# Results

## Description of sample

In total, 7,810/8,427 (93%) children and 21,508 observations were included in the pooled longitudinal analyses of $BMI_{SDS}$ and OW/OB outcomes at 8.5 years, and 6,619 in models that adjusted for pre-intervention BMI. For WC 9,718/10,028 (97%) children were included. In the longitudinal analysis of $BMI_{SDS}$ and OW/OB outcomes at 13 years, 1,533/1,907 (80%) adolescents were included, and 1,355 (71%) in models adjusted for pre-intervention BMI. Numbers excluded due to missing data were small: The largest proportion was in the 2017 cohort, where 17% were excluded due to insufficient school information to ascertain exposure status (see S3 Fig, showing the participant flow charts). Most children attended schools in urban areas in the Southern/Eastern region, reflecting the geographical distribution of the population (S2 Table). About 75% of all children attended schools in urban areas, and approximately half in the Southern/Eastern region. Approximately 20% of individuals were exposed to the FFV policy. This was higher (30%) in the 2017 cohort, reflecting oversampling in these regions. Of the 6,168 children in NFFV schools, 2,022 (33%) attended a school that had signed up to offer the parental paid FV subscription program. A full description of the cohorts is presented in S2 Table.

## Internal validity of comparisons

S2 Table shows the distribution of characteristics by attendance at a FFV or NFFV school in our sample. Children were broadly similar in terms of sex and age at outcome assessment. Differences between regions and population density were as expected, with the Northern and Central regions and less urban areas having a higher proportion of FFV schools.

Fig 1 and S3 Table compare the pre-intervention $BMI_{SDS}$ trajectories by policy exposure; similar results are shown in S4 Fig and S4 Table for the OW/OB outcome. The trajectories for $BMI_{SDS}$ and prevalence of OW/OB were broadly similar in boys; for example, with cohorts pooled, boys who would attend a FFV school had a pre-intervention $BMI_{SDS}$ 0.05 higher (95% CI: −0.06, 0.16) than those who would attend a NFFV school, after adjusting for differences in parental education, region, and population density. In girls, those who would attend a FFV school in the 2015 cohort had a more negative $BMI_{SDS}$ slope and a lower $BMI_{SDS}$ before the intervention compared to those who would attend a NFFV school. The pooled trajectories were more similar, with girls in the FFV group having a 0.08 lower pre-intervention $BMI_{SDS}$ (95% CI: −0.20, 0.034). There was little evidence for differences in the pre-intervention OW/ OB trajectory (S4 Fig; S4 Table).

## Main analysis

**Pooled.**   There was little evidence of a policy effect on $BMI_{SDS}$, OW/OB, WC, or WtHR (Fig 2) with cohorts pooled in either boys or girls at age 8.5 years, and all effect estimates were close to the null. Removing NFFV schools that offered a paid FV subscription program for most outcomes shifted effect estimates unremarkably in the direction of the null (opposite to what would be expected if the FFV policy had a causal effect; S5 Fig).

**By cohort.**   Any observed cohort-specific policy associations were inconsistent. First, among boys in the 2010 cohort, there was a suggestion of higher $BMI_{SDS}$, OW/OB, WC, and WtHR in FFV than NFFV schools (Fig 2). However, the estimates for WC and WtHR were

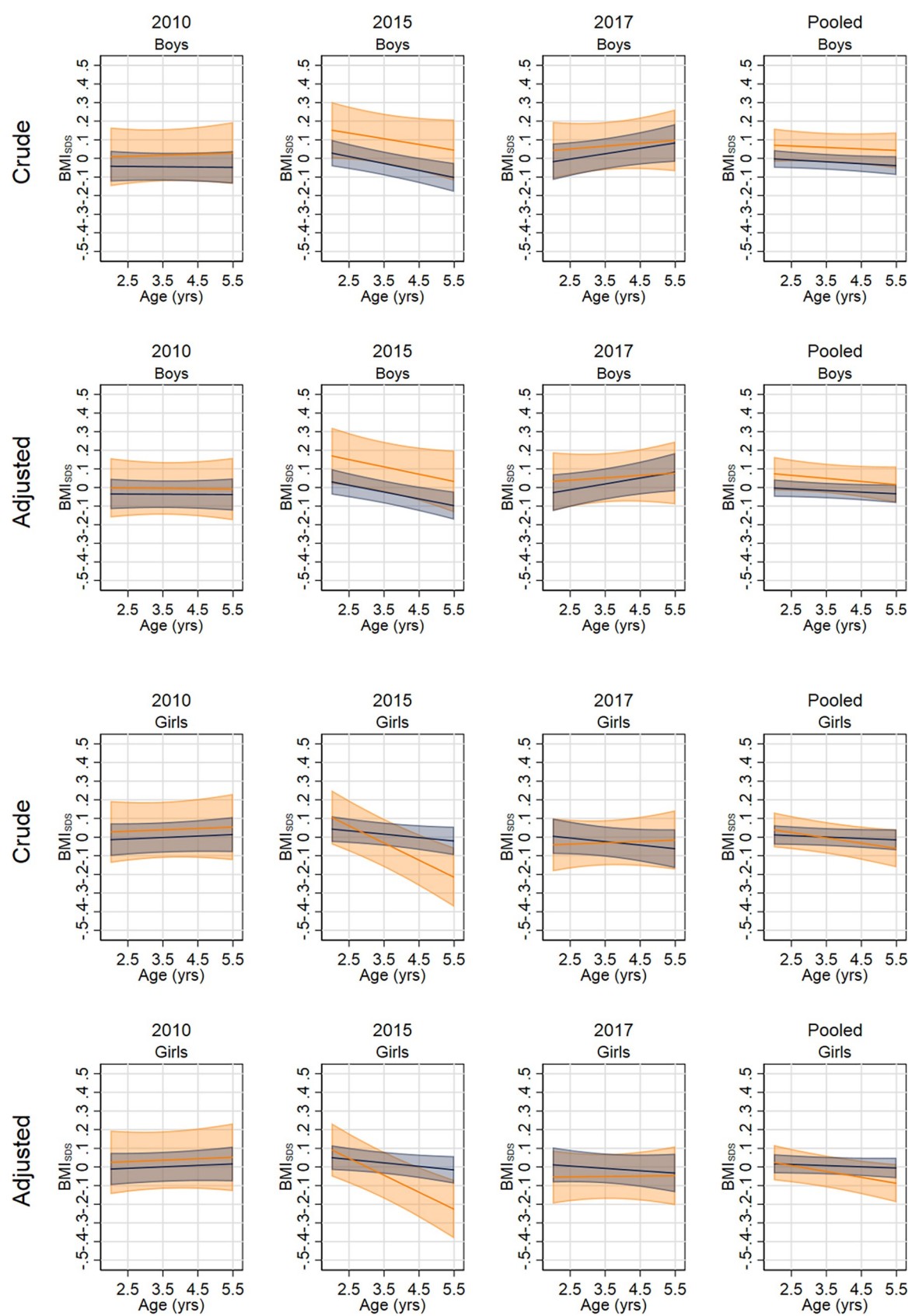

**Fig 1. Predicted pre-intervention (age 2 to 5.5 years) trajectories of $BMI_{SDS}$ in boys and girls who would attend a FFV or a NFFV school.** FFV schools (orange); NFFV schools (navy). The marginal means in each cohort and pooled cohorts and in the crude and adjusted models are presented. $BMI_{SDS}$, body mass index standard deviation score; FFV, free fruit and vegetable; NFFV, no free fruit and vegetable.

substantially attenuated after adjusting for differences in region, population density, and parental education. The estimates for the 2017 cohort ($BMI_{SDS}$, OW/OB) and 2012 cohort (WC, WtHR), which had the same exposure duration as the 2010 cohort but in which individuals were born 2 years later, were also close to the null, and so there was no replication of the 2010 suggestive findings. Removal of schools that signed up for the paid subscription program slightly increased the effect estimates in the 2010 cohort boys for $BMI_{SDS}$ and OW/OB, but slightly attenuated the estimates for WC and WtHR (S5 Fig).

Second, boys in the 2015 FFV schools, with only 1 year of FFV exposure, had a lower rather than higher $BMI_{SDS}$ (−0.12; 95% CI: −0.23, −0.01). However, this was an inconsistent dose–response pattern compared to the 2010 estimate, was attenuated after adjustment for pre-intervention $BMI_{SDS}$, and was not evident for any other outcome.

Third, girls from the same 2015 FFV schools had, on average, a higher $BMI_{SDS}$ (+0.44; 95% CI: 0.20; 0.69), but this was completely attenuated after adjusting for the differences (noted above) in pre-intervention $BMI_{SDS}$.

**By parental education.**    There was a suggestion of an interaction between the FFV policy and parental education. In the pooled and most-adjusted analyses, boys of parents without a higher education had, on average, an elevated $BMI_{SDS}$ (+0.12, $p$ for interaction = 0.04), an increased odds ratio (OR) of OW/OB (OR 1.66, $p$ for interaction = 0.02), and a higher WC (+0.7 cm, $p$ for interaction = 0.05) if they had attended a FFV school (Fig 3). This pattern was not evident in boys of parents with a higher education. The direction of this interaction was consistent across cohorts. However, the interaction was not evident for WtHR, and the interaction and effect sizes were similar or weaker after removing paid subscription schools (S6 Fig). There was also little evidence of an interaction in the girls across any outcome or cohort (Figs 3 and S8), and the direction of the interaction was in the opposite direction.

To assess whether the interaction in boys was caused by the FFV exposure or confounded by differences between school environments or the children who go to these schools, in a post hoc analysis we examined whether the same direction of interaction was evident within elementary-only schools, comparing schools that offered the paid FV subscription program versus schools that did not (see S7 Fig). We were unable to detect an interaction in these analyses, nor were interactions qualitatively in the same direction.

**Outcomes at age 13 years.**    There was little evidence for a policy effect on $BMI_{SDS}$ or OW/OB among adolescents (13 years) of either sex who had been exposed to the FFV policy for up to 4 years (Fig 4). However, there was a suggestion that girls of parents without a higher education had a lower $BMI_{SDS}$ (−0.20; 95% CI: −0.41, 0.01) and a lower odds of OW/OB (OR 0.55; 95% CI: 0.27, 1.12) if they had attended a FFV school ($p$ for both interactions = 0.05; see Fig 5) (the direction of this interaction was the same at 8.5 years but weaker). Results from the secondary analysis at age 13 years excluding NFFV schools that offered the paid FV subscription program (S8 Fig), and this analysis stratified by parental education (S9 Fig), were broadly similar.

## Population distributions

Fig 6 illustrates how the policy effect estimates from the pooled and most adjusted analyses reflect onto the population distribution of BMI and WC at 8.5 years. Shifts in the location of

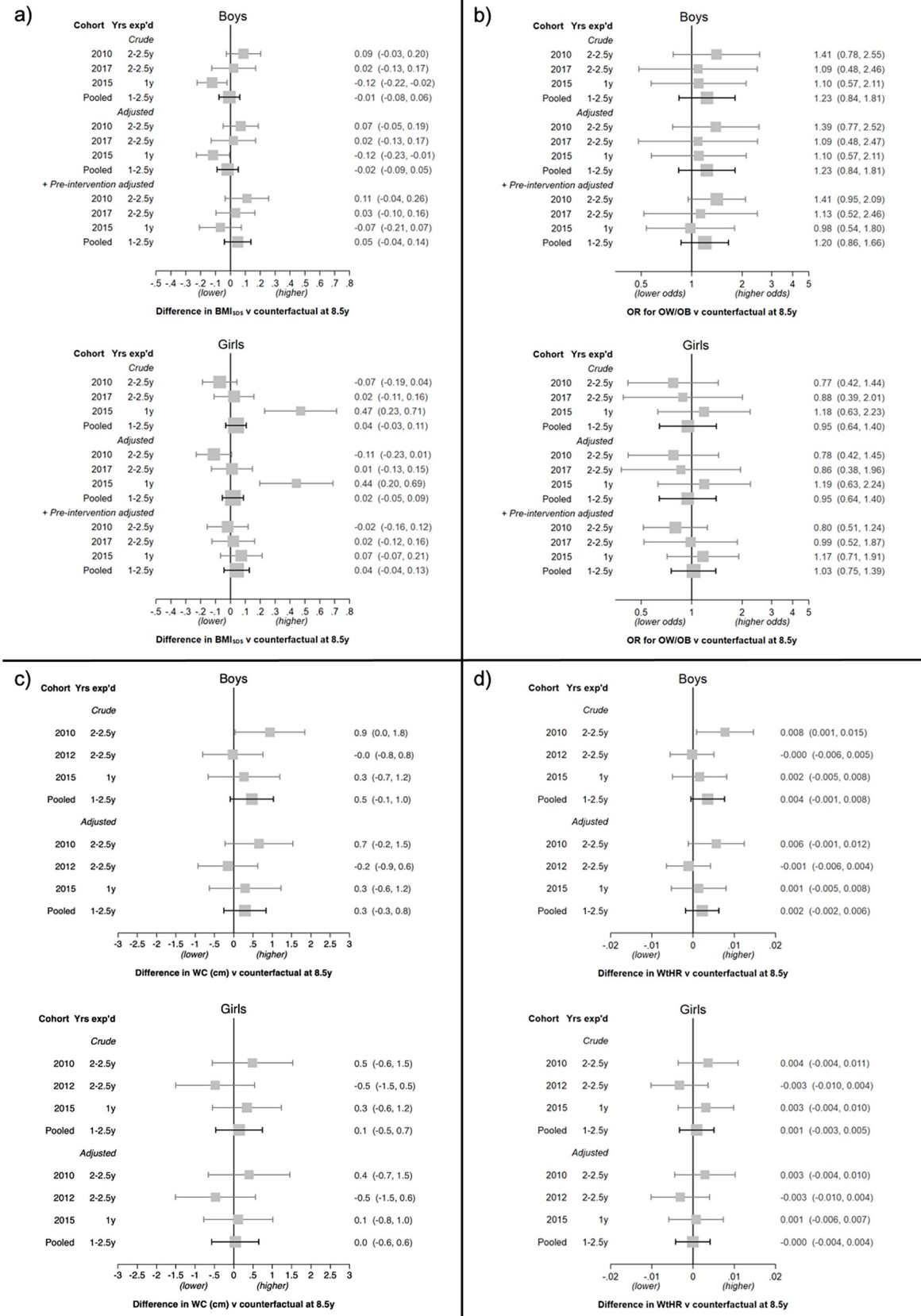

**Fig 2. Estimates of the FFV policy effect on BMI$_{SDS}$, OW/OB, WC, and WtHR at age 8.5 years.** (a) BMI$_{SDS}$; (b) OW/OB; (c) WC; (d) WtHR. Results are presented by sex and cohort (including pooled) and for each model. Expressed as the difference in outcome or OR versus the counterfactual (as estimated using the NFFV schools) with 95% CI. Analysis of BMI$_{SDS}$ and OW/OB: Pooled models include terms for cohort (intercept and slope). Adjusted models include region, population density, and highest parental education (all intercept and slope). +Pre-intervention adjusted models additionally include adjustment for BMI$_{SDS}$ prior to the intervention. Note: Pre-intervention slopes were constrained to be the same in each group for all models except for BMI$_{SDS}$ in 2015 cohort girls. Analysis of WC and WtHR: Outcomes are from grade 3 only. Pooled models include a term for cohort. Adjusted models include region, population density, and highest parental education. BMI$_{SDS}$, body mass index standard deviation score; exp'd, exposed; FFV, free fruit and vegetable; NFFV, no free fruit and vegetable; OR, odds ratio; OW/OB, overweight and obesity; WC, waist circumference; WtHR, waist to height ratio.

the distribution are small contrasted against the population variation. The bounded estimate based on the 95% CI shifted the median from a −0.07 kg/m$^2$ reduction to a +0.33 kg/m$^2$ increase. For WC this ranged from a reduction of 0.5 cm to an increase of 0.7 cm.

## Discussion

### Summary of findings

Overall, we observed little evidence that 1 to 2.5 years of exposure to a nationwide FFV policy in Norway had an appreciable benefit or unintended consequence among boys or girls with respect to childhood BMI$_{SDS}$, OW/OB, WC, or WtHR. There was some heterogeneity in the policy effect estimates in both directions at 8.5 years across cohorts, sex, and parental education although the results were inconsistent with other group comparisons, or with further adjustment for pre-policy BMI. Additionally, we observed little evidence for a policy effect at age 13 years in the cohort that had a longer duration of FFV exposure (4 years). There was a weak interaction with parental education in girls, suggesting a lower BMI$_{SDS}$ and reduced odds of OW/OB at 13 years among girls who attended FFV schools and who had parents without a higher education; however, we were unable to further test this finding in another cohort.

### Comparison with previous studies

A 2-year follow-up evaluation of a FFV program in Arkansas, US, showed a mean 0.17 *z*-score reduction in BMI among children exposed to the FFV program compared to strictly matched unexposed children, and a 3 percentage point reduction in school-level obesity as a result of the program [8]. While the confidence intervals from our pooled results overlap with their findings, we observed little evidence to support such a benefit in our sample. However, the Arkansas study was in a predominately low-income setting, reflecting a substantially different target population compared to our study. The prevalence of childhood OW/OB in Norway is approximately 16% [22] versus almost 40% in Arkansas, US [8], and children from all socio-economic positions were targeted by the Norwegian policy. Further, the matched analysis in the Arkansas study addresses a different question: It seeks the policy effect in those eligible for the intervention, while ours is concerned with the policy effect in the whole population. These factors may explain some of the differences. Our lack of observed evidence for a benefit from the FFV policy is supported by a much smaller Norwegian intervention study evaluating the association of 1 school year of FFV provision in Norwegian schools with overweight [10,19].

Findings from a meta-analysis and a systematic review of RCTs indicate beneficial effects of FV consumption on weight outcomes [11,12]; however, the interventions evaluated are heterogenous in regard to complexity, setting, and/or target populations, e.g., those with chronic conditions [11]. Moreover, studies evaluating the effect of various dietary interventions and policies on childhood obesity usually include additional components beyond FV provision [15,27–30]. Two recently published systematic reviews reported improvements in childhood

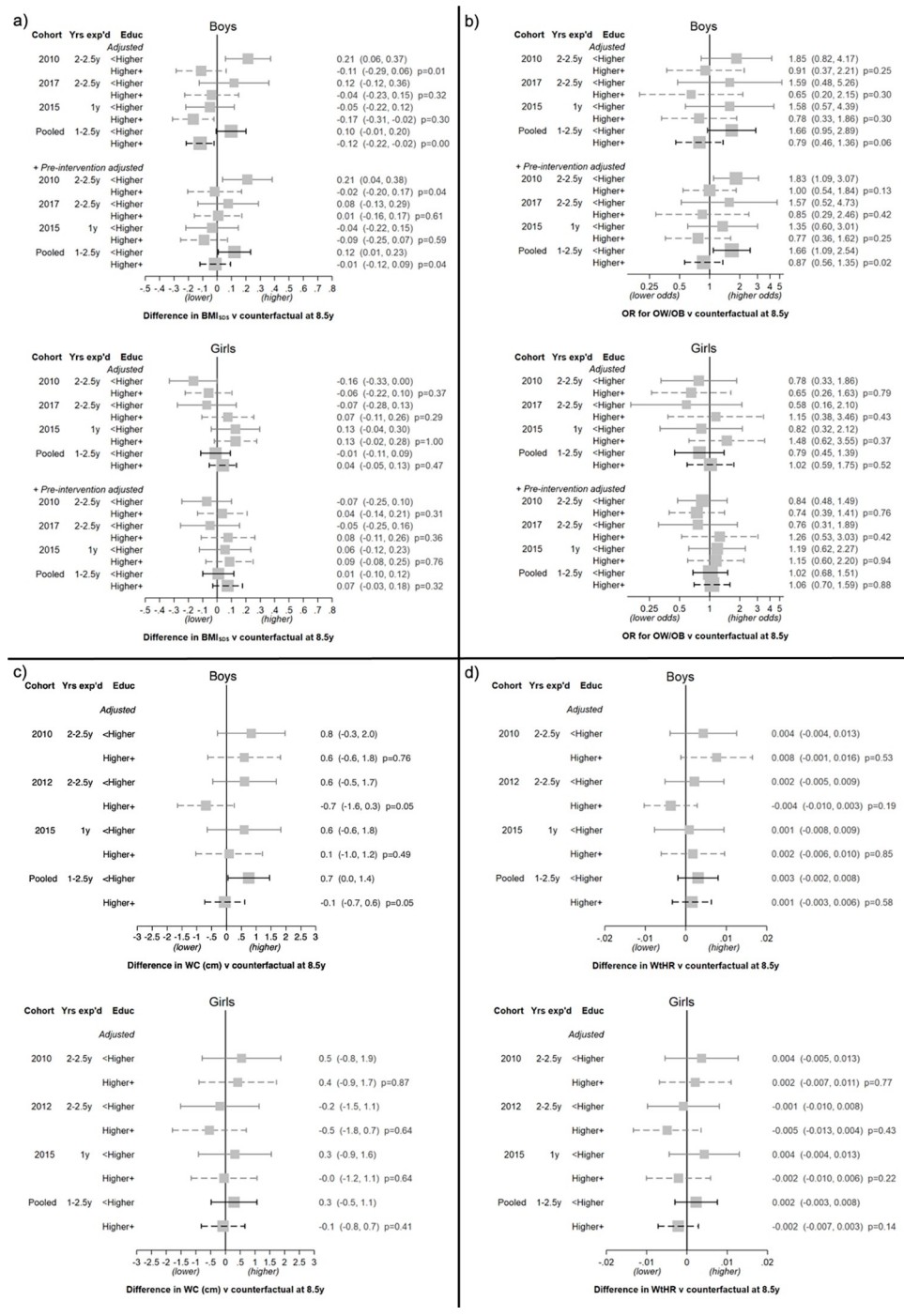

**Fig 3. Estimates of the FFV policy effect on BMI_SDS, OW/OB, WC, and WtHR at age 8.5 years, stratified by highest parental education level.** (a) BMI_SDS; (b) OW/OB; (c) WC; (d) WtHR. Results are presented by sex, cohort (including pooled), and parental education for each model. Expressed as the difference in outcome or OR versus the counterfactual (as estimated using the NFFV schools) with 95% CI. The *p*-values are from a Wald test of the interaction between parental education and FFV. Analysis of BMI_SDS and OW/OB: Pooled models include terms for cohort (intercept and slope). Adjusted models include region and population density (all intercept and slope). +Pre-intervention adjusted models additionally include adjustment for BMI_SDS prior to the intervention. Analysis of WC and WtHR: Outcomes are from grade 3 only. Pooled models include a term for cohort. Adjusted models include region and population density. BMI_SDS, body mass index standard deviation score; Educ, parental education; exp'd, exposed; FFV, free fruit and vegetable; NFFV, no free fruit and vegetable; OR, odds ratio; OW/OB, overweight and obesity; WC, waist circumference; WtHR, waist to height ratio.

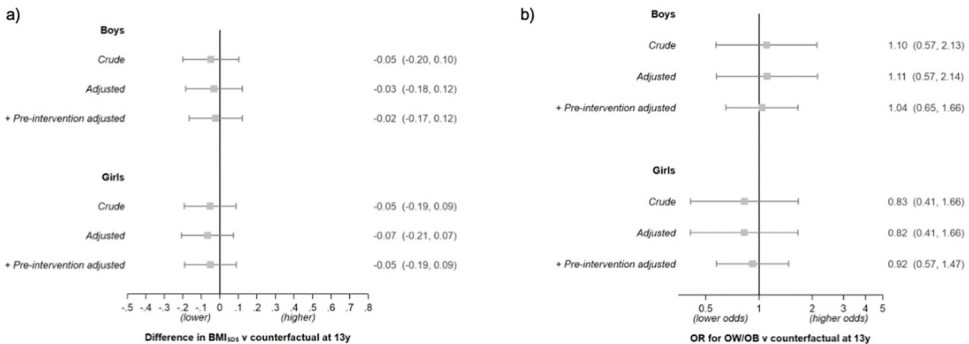

**Fig 4. Estimates of the FFV policy effect on BMI$_{SDS}$ and OW/OB at age 13 years.** (a) BMI$_{SDS}$; (b) OW/OB. Results are presented by sex for each model and expressed as the difference in outcome or OR versus the counterfactual at 13 years (as estimated using the NFFV schools) with 95% CI. Note that data are from the 2017 cohort only. Crude models have no adjustment. Adjusted models include region, population density, and highest parental education (intercept and slopes). +Pre-intervention adjusted models include additional adjustment for BMI$_{SDS}$ prior to the intervention. BMI$_{SDS}$, body mass index standard deviation score; FFV, free fruit and vegetable; NFFV, no free fruit and vegetable; OR, odds ratio; OW/OB, overweight and obesity.

BMI from school food environment interventions focusing on competitive food and beverage policies [29] and using clear and concise dietary guidelines [28], indicating that complex interventions and/or policies may benefit childhood obesity. Altogether, these studies include aspects that are beyond comparison to a nationwide FFV policy, which make them sufficiently different to be used as part of the evidence base to inform a FFV policy implementation compared to our study.

## Interpretations

One explanation for the absence of a clear beneficial effect of the Norwegian FFV policy may be that exposed children did not substitute higher energy foods, such as unhealthy snacks, with FVs, which has previously been proposed as a possible pathway for weight loss [14,31]. This possibility is supported by findings reported after the first year of the Norwegian FFV

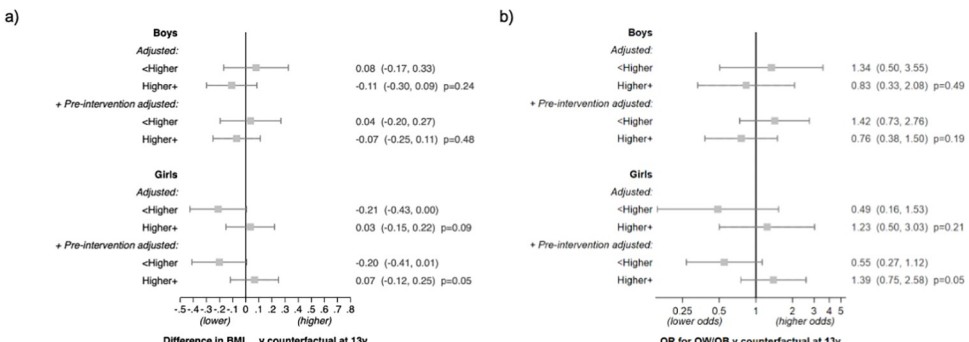

**Fig 5. Estimates of the FFV policy effect on BMI$_{SDS}$ and OW/OB at age 13 years stratified by highest parental education level.** (a) BMI$_{SDS}$; (b) OW/OB. Results are presented by sex and parental education for each model. Expressed as the difference in outcome or OR versus the counterfactual (as estimated using the NFFV schools) with 95% CI. The *p*-values are from a Wald test of the interaction between parental education and FFV. Note that data are from the 2017 cohort only. Adjusted models include terms for region and population density (intercept and slopes). +Pre-intervention adjusted models include additional adjustment for BMI$_{SDS}$ prior to the intervention. BMI$_{SDS}$, body mass index standard deviation score; FFV, free fruit and vegetable; NFFV, no free fruit and vegetable; OR, odds ratio; OW/OB, overweight and obesity.

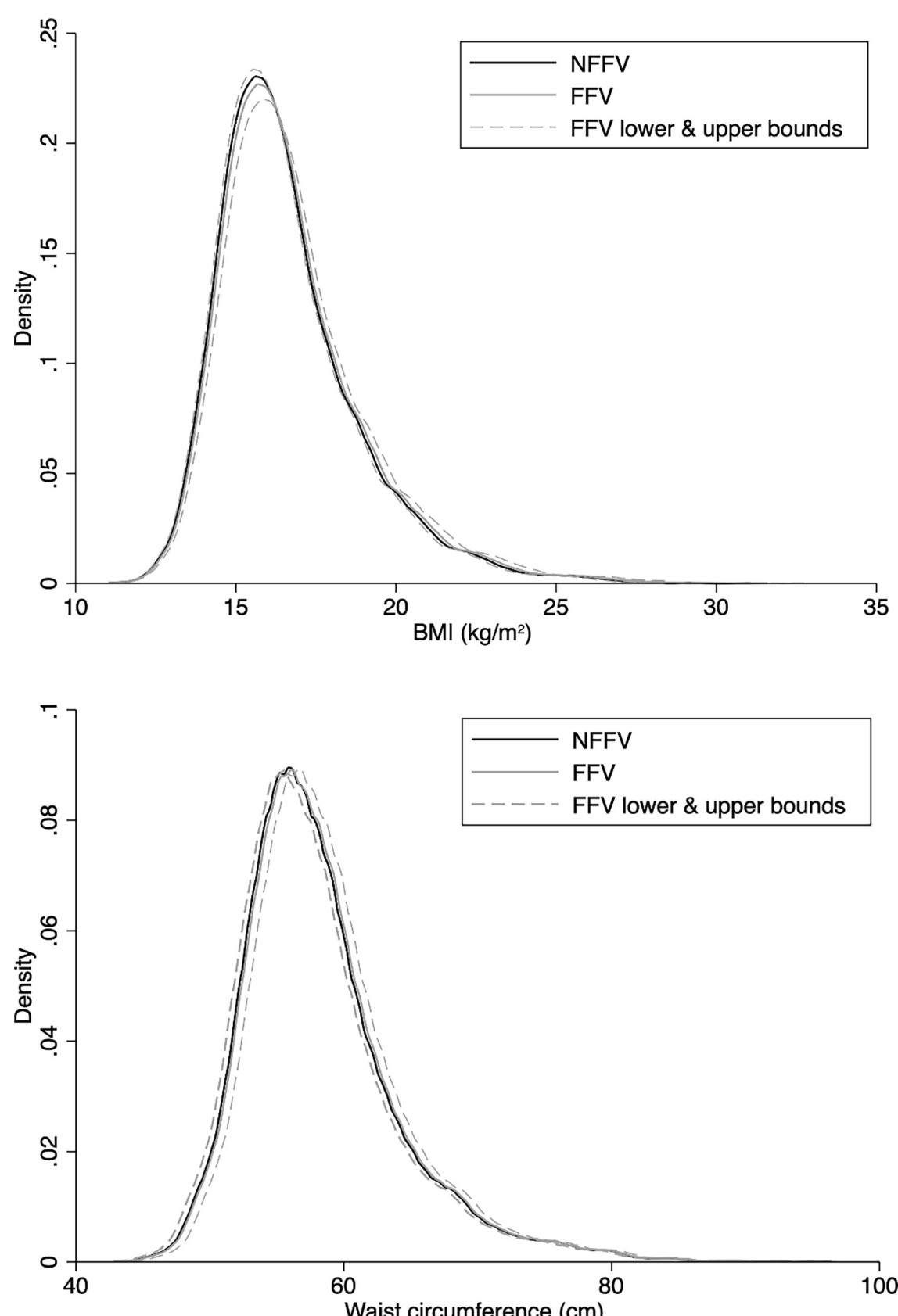

**Fig 6. Model-based predictions for the FFV policy effect on the distribution of BMI (kg/m$^2$) and waist circumference (cm) at 8.5 years.** Estimates use the point estimates and 95% confidence intervals to give a bounded prediction for the FFV effect. The estimates are from the +pre-intervention adjusted models in boys and girls. A kernel density smoother was used to illustrate the distribution. BMI, body mass index; FFV, free fruit and vegetable; NFFV, no free fruit and vegetable.

policy indicating no substantial differences in the consumption of unhealthy energy-dense snacks, despite an increased odds of daily fruit consumption among adolescents (mean age 14.5 years) attending FFV schools compared to those attending NFFV schools [32]. On the other hand, when solely adding daily FVs to the diet without any compensatory behavior changes (e.g., eating less of other foods or increasing physical activity level), one might expect an increase in weight outcomes. However, FVs are generally low in energy, and providing 1 portion of fresh FVs each school day may not contribute to an excessive energy intake. Substitution and compensatory behavior changes in response to the FFV policy among some children but not others might result in no overall aggregated policy effect in the population, as suggested by our pooled estimates.

We anticipated confounding to act in the direction of weight gain due to the predominance of FFV schools in less population-dense areas that have slightly higher levels of OW/OB [22]. If results were biased in this direction, as for the most part our results suggest, it is reassuring that there was still no consistent evidence of unintended consequences from the FFV policy. Further, our upper bound prediction of the policy's effect on the population distribution of BMI and WC would suggest that even in the worst-case scenario, a FFV policy is probably unlikely to cause a population shift of concern. Nonetheless, it should be mentioned that our stratified analysis showed an interaction of the FFV policy and parental education among boys suggesting an increased BMI$_{SDS}$ and odds of OW/OB among boys of parents without higher education exposed to the FFV policy compared to those unexposed. This result was driven by the earliest born (2010) cohort. While healthier behavior patterns and changes to the obesogenic environment over time may explain this (see examples in Table A in S7 Text), the inconsistency of this result with our other comparisons and with our secondary analysis suggest chance or confounding as the most plausible explanation.

In the present study, even with the relatively large sample of 1,533 adolescents in the 2017 cohort who were exposed to the FFV policy for up to 4 years, few consistent reductions in weight outcomes were observed. The lack of observed associations with weight status may partly reflect the repeal of the FFV policy in 2014, meaning that, at the time of the 13-year measurement, 3 years had passed since FFV provision in school. However, analysis stratified by parental education among adolescents in the 2017 cohort indicated lower BMI$_{SDS}$ and reduced odds of OW/OB among girls who attended FFV schools and who had parents without higher education, compared to unexposed girls. Norwegian girls generally report eating more fruit and berries than boys [33]. Additionally, a sufficiently long follow-up period could be of importance to detect possible effects on body weight from a FFV policy [34], which might explain this beneficial finding among girls of parents without higher education. Another Norwegian study reported significantly higher sustained fruit consumption among less-educated young women who in childhood had received 1 school year of FFV compared to controls [35]. Nonetheless, this result should be interpreted with caution and requires replication.

## Implications and further work

FFV policies and programs have been shown to increase consumption of FVs [6,36] and may thereby improve nutrient intake and other health outcomes [37]. However, our findings question whether FFV policies and programs alone can be expected to reduce rates of childhood or

adolescent OW/OB when causes of obesity are multifaceted [38]. One or 2 of the interactions between weight outcomes and parental education require further investigation, and we recommend that future studies that investigate nationwide policies should be population-wide and sufficiently powered to assess heterogeneity across boys and girls from different socioeconomic positions and across other more vulnerable subgroups. Studies should also be sufficiently large to detect small but potentially meaningful population-level effects on OW/OB outcomes. Including data on additional variables such as attitudes, values, and FV consumption at the individual level may aid the understanding of potential mechanisms of how FFV policies act. Additionally, as provision of FVs may contribute to promoting healthy eating habits, future work should evaluate whether a FFV policy contributes to longer-term healthy eating habits and thereby prevents OW/OB in adulthood [12].

## Strengths and limitations

Although our study was nationwide, generalizability might be limited to countries with a similar prevalence of OW/OB [39]. The use of longitudinal data in the current study allowed the assessment of pre-intervention weight trajectories and the construction of a more plausible counterfactual to estimate the policy effect compared to difference-in-difference or cross-sectional designs used in similar previous evaluations [8,10]. The high-quality objective data, which were standardized and cleaned using a systematic approach [23], and the use of models that made use of all available outcome measures and handled the relatively small amount of missingness in a principled way, are also strengths. Further, we were also able to look at WC as an outcome, acknowledging that BMI has limitations as a marker of excess adiposity among children [40]. However, our sample size was insufficient to allow us to assess effects on obesity (BMI $\geq$ 30 kg/m$^2$), which has a relatively low prevalence in Norwegian children [22]. We also lacked information on consumption of the FFVs that may have enhanced interpretation and translation of our findings.

The lack of a pre-registered protocol for our study may undermine findings even though little evidence for a policy effect was observed. Using the ROBINS-I tool [41], we assessed the potential overall risk of bias in our study to be moderate (details in S8 Text). Since we were unable to assume "as if" random allocation of the FFV policy, residual confounding is a key risk of bias, as is misclassification of exposure caused by some children attending both a FFV and NFFV school. However, the slopes of the pre-policy trajectories were for the most part quite similar, and the use of multiple cohorts and additional school information allowed us to draw stronger conclusions by assessing the consistency of the evidence from several sets of comparisons, each with the potential for different biases. A list of these comparisons, the secondary and sensitivity analyses that were done to check the robustness and consistency of results, and an assessment of potential biases are provided in S5 Table. The risk of bias due to other co-interventions was deemed low (see S7 Text), and checks of the robustness of the results to the choice of analysis strategy suggest that this was probably unlikely to have influenced our key findings (see S5 Table). There is inevitable bias compared to a well-controlled RCT; however, we do not predict this bias to be sufficient to alter our main conclusions.

## Conclusion

We observed little evidence that exposure to a nationwide FFV policy had any notable beneficial effect or unintended consequence on weight status among Norwegian children and adolescents. While a nationwide FFV policy alone is unlikely to have a substantial impact on population childhood weight outcomes, given the benefits linked to enhanced nutrition, as documented in other studies, a national policy may have benefits for other aspects of health

and dietary behavior without the unintended consequences that are a risk of such population-wide interventions.

## Supporting information

**S1 Checklist. STROBE checklist of items included in "A nationwide school fruit and vegetable policy and childhood and adolescent overweight: A quasi-natural experimental study."**
(DOCX)

**S1 Fig. Schematic of the quasi-natural experimental design.** The dashed square indicates the period with the FFV policy; the squares indicate measurements in the NCGS (2010, 2012, and 2015) and NYGS (2017); and the dots indicate approximate (routine) measurements included in analysis. FFV, free fruit and vegetable; NCGS, Norwegian Childhood Growth Study; NYGS, Norwegian Youth Growth Study.
(DOCX)

**S2 Fig. Plot of individual values used in the analysis samples of BMI in each cohort (2010, orange; 2015, green; 2017, brown).** BMI, body mass index.
(DOCX)

**S3 Fig. Participant flow charts by cohort.** [*]Lost individuals are missing outcome. [†]Pre-intervention BMI adjusted model. Adj, adjusted; BMI, body mass index; Educ, parental education; FFV, free fruit and vegetable; NFFV, no free fruit and vegetable; pop-den, population density; WC, waist circumference.
(DOCX)

**S4 Fig. Predicted pre-intervention (2 to 5.5 years) trajectories of overweight (including obesity) in boys and girls who would attend a FFV (orange) versus a NFFV school (navy).** The marginal proportions in each cohort and pooled cohorts, and in the crude and adjusted models, are presented. FFV, free fruit and vegetable; NFFV, no free fruit and vegetable.
(DOCX)

**S5 Fig. Secondary analysis showing estimates of the FFV policy effect excluding NFFV schools that took part in the parental paid subscription program on $BMI_{SDS}$, OW/OB, WC, and WtHR at 8.5 years.** (a) $BMI_{SDS}$; (b) OW/OB; (c) WC; (d) WtHR. Results are presented by sex and cohort (including pooled) and for each model. Expressed as the difference in outcome or OR versus the counterfactual (as estimated using the NFFV schools) with 95% CI. Analysis of $BMI_{SDS}$ and OW/OB: Pooled models include terms for cohort (intercept and slope). Adjusted models include region, population density, and highest parental education (all intercept and slope). +Pre-intervention adjusted models additionally include adjustment for $BMI_{SDS}$ prior to the intervention. Note: Pre-intervention slopes were constrained to be the same in each group for all models except for $BMI_{SDS}$ in 2015 cohort girls. Analysis of WC and WtHR: Outcomes are from grade 3 only. Pooled models include a term for cohort. Adjusted models include region, population density, and highest parental education. $BMI_{SDS}$, body mass index standard deviation score; FFV, free fruit and vegetable; NFFV, no free fruit and vegetable; OR, odds ratio; OW/OB, overweight and obesity; WC, waist circumference; WtHR, waist to height ratio.
(DOCX)

**S6 Fig. Secondary analysis showing estimates of the FFV policy effect excluding NFFV schools that took part in the parental paid subscription program on $BMI_{SDS}$, OW/OB,**

**WC, and WtHR at 8.5 years, stratified by highest parental education level.** (a) BMI$_{SDS}$; (b) OW/OB; (c) WC; (d) WtHR. Results are presented by sex, cohort (including pooled), and parental education for each model. Expressed as the difference in outcome or OR versus the counterfactual (as estimated using the NFFV schools) with 95% CI. The *p*-values are from a Wald test of the interaction between parental education and FFV. Analysis of BMI$_{SDS}$ and OW/OB: Pooled models include terms for cohort (intercept and slope). Adjusted models include region and population density (all intercept and slope). +Pre-intervention adjusted models additionally include adjustment for BMI$_{SDS}$ prior to the intervention. Analysis of WC and WtHR: Outcomes are from grade 3 only. Pooled models include a term for cohort. Adjusted models include region and population density. BMI$_{SDS}$, body mass index standard deviation score; FFV, free fruit and vegetable; NFFV, no free fruit and vegetable; OR, odds ratio; OW/OB, overweight and obesity; WC, waist circumference; WtHR, waist to height ratio. (DOCX)

**S7 Fig. Estimates of the school fruit and vegetable subscription program (paid versus not paid) on BMI$_{SDS}$, OW/OB, WC, and WtHR at 8.5 years, stratified by highest parental education level.** (a) BMI$_{SDS}$; (b) OW/OB; (c) WC; (d) WtHR. Results are presented by cohort (including pooled) and parental education for each model. Expressed as the difference in outcome or OR versus the counterfactual (as estimated using the NFFV schools without the subscription program) with 95% CI. The *p*-values are from a Wald test of the interaction between parental education and the subscription program. Analysis of BMI$_{SDS}$ and OW/OB: Pooled models include terms for cohort (intercept and slope). Adjusted models include region and population density (all intercept and slope). +Pre-intervention adjusted models additionally include adjustment for BMI$_{SDS}$ prior to the intervention. Analysis of WC and WtHR: Outcomes are from grade 3 only. Pooled models include a term for cohort. Adjusted models include region and population density. BMI$_{SDS}$, body mass index standard deviation score; NFFV, no free fruit and vegetable; OR, odds ratio; OW/OB, overweight and obesity; WC, waist circumference; WtHR, waist to height ratio. (DOCX)

**S8 Fig. Secondary analysis showing estimates of the FFV policy effect excluding NFFV schools that took part in the parental paid subscription program on BMI$_{SDS}$ and OW/OB at age 13 years.** (a) BMI$_{SDS}$; (b) OW/OB. Results are presented by sex for each model and expressed as the difference in outcome or OR versus the counterfactual at 13 years (as estimated using the NFFV schools) with 95% CI. Note that data are from the 2017 cohort only. Crude model has no adjustment. Adjusted models include region, population density, and highest parental education (intercept and slopes); +Pre-intervention adjusted models include additional adjustment for BMI$_{SDS}$ prior to the intervention. BMI$_{SDS}$, body mass index standard deviation score; FFV, free fruit and vegetable; NFFV, no free fruit and vegetable; OR, odds ratio; OW/OB, overweight and obesity. (DOCX)

**S9 Fig. Secondary analysis showing estimates of the FFV policy effect excluding NFFV schools that took part in the parental paid subscription program on BMI$_{SDS}$ and OW/OB at age 13 years, stratified by highest parental education level.** (a) BMI$_{SDS}$; (b) OW/OB. Results are presented by sex and parental education for each model. Expressed as the difference in outcome or OR versus the counterfactual (as estimated using the NFFV schools) with 95% CI. The *p*-values are from a Wald test of the interaction between parental education and FFV. Note that data are from the 2017 cohort only. Adjusted models include terms for region and population density (intercept and slopes). +Pre-intervention adjusted models include

additional adjustment for BMI$_{SDS}$ prior to the intervention. BMI$_{SDS}$, body mass index standard deviation score; FFV, free fruit and vegetable; NFFV, no free fruit and vegetable; OR, odds ratio; OW/OB, overweight and obesity.
(DOCX)

**S1 Table. Frequencies of schools, children, and observations by county illustrating the hierarchical data structure of the 3 longitudinal cohorts (pooled) based on the analysis sample.** [†]Region and county at recruitment. FFV, free fruit and vegetable; NFFV, no free fruit and vegetable; Obs, observations.
(DOCX)

**S2 Table. Description of individuals included in the analysis of outcomes at age 8.5 (third grade) by attendance at a FFV school in each cohort and pooled across cohorts.** [*]NFFV: Individuals who did not attend a school with FFV provision. FFV ≥ 1 year: Individuals who attended a school with FFV provision at least 1 year. [‡]In third grade. [†]Of individuals attending NFFV schools, proportion who attended a school offering the paid fruit and vegetable sub-scription program. [§]Parental education prior to possible exposure (when the child was 4 years old). [a]These cohorts had longitudinal data and were pooled in the analysis of BMI$_{SDS}$ and OW/OB. BMI$_{SDS}$, body mass index standard deviation score; FFV, free fruit and vegetable; NA, not applicable; NFFV, no free fruit and vegetable; OW/OB, overweight and obesity; Paid-sub, individuals attending schools offering the parental paid subscription program; SD, standard deviation.
(DOCX)

**S3 Table. Estimated differences in pre-intervention (2 to 5.5 years) trajectories of BMI$_{SDS}$ in boys and girls who would attend a FFV versus a NFFV school.** Differences in slope from 2 to 5.5 years and in BMI$_{SDS}$ at 5.5 years in each cohort and pooled cohorts, and in the crude and adjusted models, are presented. [†]Crude pooled models include adjustment for cohort (intercept and slope). All models include a random intercept for school and random coefficients for child. [‡]Adjusted models include region, population density, and highest parental education (intercept and slope); pooled adjusted models also include terms for cohort (intercept and slope). All models include random intercepts for school and random coefficients for child. [a]Difference in slope (BMI$_{SDS}$ per year): FFV minus NFFV. [b]Difference in BMI$_{SDS}$ at 5.5 years: FFV minus NFFV. BMI$_{SDS}$, body mass index standard deviation score; FFV, free fruit and vegetable; NFFV, no free fruit and vegetable.
(DOCX)

**S4 Table. Odds ratios comparing pre-intervention (age 2 to 5.5 years) trajectories of over-weight including obesity in boys and girls who would attend a FFV versus a NFFV school.** The ORs compare the slopes of the log odds of OW/OB from age 2 to 5.5 years and the odds of OW/OB at age 5.5 years (pre-intervention age). [†]Crude pooled models include adjustment for cohort (intercept and slope). All models include a random intercept for school and child. [‡]Adjusted models include region, population density, and highest parental education (inter-cept and slope); pooled adjusted models also include terms for cohort (intercept and slope). All models include random intercepts for school and child. [a]OR comparing slopes of log odds (log odds per year) of overweight: FFV/NFFV. [b]OR comparing log odds of overweight at 5.5 years (pre-intervention): FFV/NFFV. FFV, free fruit and vegetable; NFFV, no free fruit and vegetable; OR, odds ratio.
(DOCX)

**S5 Table. Summary of some of the analyses that were performed to check robustness and consistency of results.**
(DOCX)

**S1 Text. Standardizing the BMI outcome (BMI standard deviation scores).**
(DOCX)

**S2 Text. Exposure to FFV policy classification.**
(DOCX)

**S3 Text. Longitudinal estimation of the FFV policy effect.**
(DOCX)

**S4 Text. Regional patterning of combined elementary and secondary (FFV) and elementary-only schools (NFFV).**
(DOCX)

**S5 Text. Directed acyclic graph.**
(DOCX)

**S6 Text. Sensitivity analysis with different classifications of parental education.**
(DOCX)

**S7 Text. National policy initiatives and co-interventions occurring over the time frame of the study.**
(DOCX)

**S8 Text. ROBINS-I tool for risk of bias in non-randomized comparisons.**
(DOCX)

## Acknowledgments

Jørgen Meisfjord contributed with expertise regarding sampling, and Ingvild Bokn oversaw data collection of the NYGS. Tore Angelsen at the Norwegian Fruit and Vegetable Marketing Board provided valuable information regarding provision of FVs in schools. We are particularly grateful to the school health nurses for collection of data and to all the participants.

## Author Contributions

**Conceptualization:** Tonje H. Stea, Elling Bere, Per Magnus, Andrew K. Wills.

**Formal analysis:** Bente Øvrebø, Andrew K. Wills.

**Methodology:** Elling Bere, Per Magnus, Andrew K. Wills.

**Visualization:** Bente Øvrebø, Andrew K. Wills.

**Writing – original draft:** Bente Øvrebø, Andrew K. Wills.

**Writing – review & editing:** Bente Øvrebø, Tonje H. Stea, Ingunn H. Bergh, Elling Bere, Pål Surén, Per Magnus, Petur B. Juliusson, Andrew K. Wills.

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
