## [Editor Report · Decision Letter 0]

22 Sep 2021

Dear Dr Øvrebø, 

Thank you for submitting your manuscript entitled "No strong evidence of benefits or unintended consequences on BMI or overweight from a nationwide school fruit and vegetable policy: a quasi-natural experimental study" for consideration by PLOS Medicine.

Your manuscript has now been evaluated by the PLOS Medicine editorial staff as well as by an academic editor with relevant expertise and I am writing to let you know that we would like to send your submission out for external peer review.

Please re-submit your manuscript within two working days, i.e. by Sep 24 2021 11:59PM.

Kind regards,

Callam Davidson

Associate Editor

PLOS Medicine

---

## [Decision Letter · Decision Letter 1]

15 Oct 2021

Dear Dr. Øvrebø,

Thank you very much for submitting your manuscript "No strong evidence of benefits or unintended consequences on BMI or overweight from a nationwide school fruit and vegetable policy: a quasi-natural experimental study" (PMEDICINE-D-21-04001R1) for consideration at PLOS Medicine. 

Your paper was evaluated by an associate editor and discussed among all the editors here. It was also discussed with an academic editor with relevant expertise, and sent to independent reviewers, including a statistical reviewer. The reviews are appended at the bottom of this email and any accompanying reviewer attachments can be seen via the link below:

[LINK]

In light of these reviews, I am afraid that we will not be able to accept the manuscript for publication in the journal in its current form, but we would like to consider a revised version that addresses the reviewers' and editors' comments. We cannot make any decision about publication until we have seen the revised manuscript and your response, and we plan to seek re-review by one or more of the reviewers. 

We hope to receive your revised manuscript by Nov 05 2021 11:59PM. Please email us (plosmedicine@plos.org) if you have any questions or concerns.

We look forward to receiving your revised manuscript. 

Sincerely,

Callam Davidson,

Associate Editor 

PLOS Medicine

plosmedicine.org

Please revise your title according to PLOS Medicine's style. Your title must be nondeclarative. It should begin with main concept if possible. "Effect of" should be used only if causality can be inferred, i.e., for an RCT. Please place the study design in the subtitle (ie, after a colon). See PLOS Medicine's website for examples: https://journals.plos.org/plosmedicine/

Please update your Data Availability Statement to remove ‘for non-Norwegians’, as the URL provided appears to contain information aimed at both those that do and those that do not hold a Norwegian electronic ID (if I have understood correctly). 

In the Abstract Methods and Findings please include the important dependent variables that are adjusted for in the main analyses.

Please begin the last sentence of the Abstract Methods and Findings section: ‘The main limitations of this study are…’.

In your Abstract Conclusions, please address the study implications without overreaching what can be concluded from the data; the phrase "In this study, we observed ..." may be useful, given that the study has an observational design.

Please confirm that informed consent (provided by parents/guardians) was written (rather than oral) for all of the involved cohorts (the current manuscript suggests this was the case but it would be useful to have a short sentence confirming).

Thank you for including the completed STROBE checklist as Supporting Information. Please add the following statement, or similar, to the Methods: "This study is reported as per the Strengthening the Reporting of Observational Studies in Epidemiology (STROBE) guideline (S1 Checklist)."

Please update the STROBE checklist to use section and paragraph numbers, rather than page numbers.

Did your study have a prospective protocol or analysis plan? Please state this (either way) early in the Methods section.

Please remove the Authors’ contributions, Data availability, Funding, and Conflicts of Interest sections from the end of the main text. In the event of publication, this information will be published as metadata based on your responses to the relevant questions in the submission form.

Please relocate your Ethics approval section from the end of the main text to the Methods.

Comments from the reviewers:

Reviewer #1: The authors estimate the impact of a policy to provide free fruit and vegetables (FFV) at school meals in Norway on outcomes related to adiposity and weight distribution. FFV programs were nonrandomly allocated and for causal inference the authors use longitudinal observational methods with correction for some potential confounding variables. These methods do not detect an impact of FFV policy exposure on BMI, odds of OW/OB, or waist circumference. I feel strongly that it is important to publish null findings, so I have no concerns about that. I am only superficially familiar with the methods used in this analysis and so will rely on the statistical reviewer for a more detailed review in that regard.

1) My main constructive critique is that the analysis appears not to exclude the possibility of meaningful effects on the odds of OB/OW, especially in the cohort with longest exposure where the estimate appears imprecise. In general, the estimates do not appear precise enough to exclude meaningful effects. I think this is something that could be mostly addressed via revision of the text. It is rarely possible to demonstrate a true null; what is possible is to exclude effect sizes of a certain magnitude. I think the paper should acknowledge that the analysis does not exclude potentially meaningful effect sizes and is not necessarily a failure to replicate previous positive findings (in the sense that 95% CIs overlap considerably), and address this point quantitatively. What size effects can be reasonably excluded by this analysis, and what size effects cannot be excluded? Discussion of this, and the uncertainty of interpretation it implies, would be helpful.

2) My second constructive critique is that I am not as convinced as the authors that the conclusions are at a low risk of being impacted by bias, as reflected in the abstract: "Residual confounding and exposure misclassification are the main threats to the validity of our estimates but are unlikely to be of sufficient magnitude to affect our conclusions". Measurement and categorization of potential confounding variables tends to be crude (e.g., dichotomizing educational status), and the number of potential confounding variables included is limited (e.g., economic status is not directly considered). I'm not knowledgeable enough about the methods used in this study to form a strong opinion on this, but based on the information presented and my general familiarity with this class of methods, I'll say I'm skeptical of the assertion that the risk of bias is low. For example, the ORs for the pre-intervention trajectories of OB/OW in table S7 have confidence intervals that include the possibility of substantial differences (e.g., 0.75 to 1.20), suggesting that it is possible that OB/OW risk trajectories in the intervention children were not well matched with controls. Perhaps the statistical reviewer can contribute an opinion on this, but my opinion is that the assertions of low risk of bias should be removed or softened.

More detailed comments, including some related to the above, are below:

3) I see no evidence that the study was preregistered. This increases risk of bias. While I am less concerned about this due to the null primary outcome, it does undermine the internal validity of the findings.

4) The analyses included 6619 to 7810 children, and only a minority of those were in the FFV group (21% in table 1). How much power did this analysis have to detect significant and quantitatively meaningful effects on BMI and OB/OW? What is the likelihood that the finding is null because the sample size is too small? I would expect that if this intervention is effective, the effect size would be quite small. I wouldn't expect offering F/V to children at school to have a large impact on BMI. I see that the 95% confidence intervals around the estimated odds ratios for OB/OW are quite wide, for example in figure 2b and especially figure 4b. This raises the question of whether the underlying data are up to the task of detecting a plausible (presumably small) effect size.

5) This is not a critique, just a comment. The theory of change for this intervention leaves several opportunities for the program to fail to impact OB/OW. Policy enacted -> F/V provided -> F/V consumed -> F/V displace more energy-dense foods -> total energy intake declines -> lower BMI. I don't find it surprising that effect size on OB would be small or zero.

6) It is a significant weakness that the data do not include a measure of the impact of the intervention on F/V consumption. This isn't strictly required since the objective of the study is to evaluate the policy, but it does complicate interpretation. I think this should be mentioned more prominently, perhaps in the abstract. 

7) "Parental education was used as an indicator for socioeconomic position. We used the highest parental education (from either mother or father) prior to the policy exposure when the children were four years old." This seems like a crude measure of SES, especially since the measure was binary (higher+ vs. <higher). It seems like the measure of geographic location may also be crude, although I don't know enough about Norway to assess this with confidence. Only four locations were considered: North; Mid; West; and South-East. I would expect substantial variation in population characteristics by location within these large geographic regions. Categorization of populations into urban, semi-urban, and rural seems reasonable.

8) Concerns about possible confounding are heightened by the following: "Allocation of the FFV policy could not be considered 'as if' random. Combined (FFV) schools are more likely to be in areas of lower population density compared to pure elementary (NFFV) schools and are thus more common in rural regions of Norway such as the North (see S4 Text and S5 Table)" Table 1 also indicates differences in region and education level between FFV and NFFV groups. The authors adjust for measured confounders but clearly the populations being compared are not the same. The authors acknowledge this and attempt to adjust for it, but I think they are perhaps too confident that the conclusions are not affected by it, given the limitations of how possible confounders were measured and categorized.

9) In figure 1, I believe the bottom two pooled graphs are mislabeled as boys.

10) Figure 6 is nice. I like seeing the full distributions. It's somewhat hard to see the lines though.

11) The discussion says (509-512) "A two-year follow-up evaluation of the FV program in Arkansas US, showed a mean 0.17 z-score reduction in BMI among children exposed to the FFV program compared to strictly matched unexposed children, and also a three percentage point reduction in school-level obesity as a result of the program [8]. We found no evidence to support such a benefit." This is accurate, but unless I misunderstood something, the current study may not have been able to detect an effect size of this magnitude (especially the older cohort with longer exposure). If baseline OB/OW prevalence is 40% and F/V reduced it to 37%, that's an OR of 0.88 ((40/60) / (37/63)), which appears to be within or close to the 95% CIs of the current study. So it may be misleading to frame this as a failure of replication. I think this is doubly true if one considers that the 95% CI of the two estimates probably overlap quite a bit. This point should be discussed. I see that in the Arkansas study, the sample size of the intervention group was not especially large, but they went to greater lengths to match intervention and control subjects. This may have yielded greater power to detect a smaller effect.

Thank you,

Stephan J Guyenet

Reviewer #2: Manuscript PMEDICINE-D-21-04001R1 with title "No strong evidence of benefits or unintended consequences on BMI or overweight from a nationwide school fruit and vegetable policy: a quasi-natural experimental study" provides an interesting study on secular trends of overweight among adolescents in Norway and explores whether a free fruit program had any effects. Authors applied very sophisticated and robust modelling for their study. Basically their findings show that having a national policy on school fruit and vegetables does not affect BMI. It is good practice to publish non significant results of public health nutrition interventions.

What I miss from this paper is a more thorough multidisciplinary discussion. It is a pity that data is already collected, and I am not asking for this, however, it is key to note that it would have benefited from more variables e.g. attitudes or values. Results are not well explained/discussed, although they could have been better explained if e.g. focus groups or interviews would have been performed, or additional consumer-relevant information would have been collected together with the food intake and anthropometric measurements. The confounding in such studies is large, and authors do not mention what kind of recommendations are usual at that age, or the determinants of food choices among adolescents, or the environmental cues in Norway. Have there been any behaviour interventions accompanying the availability change? Was there any special recommendation or social marketing campaigns to support the F&V intake? Authors have not addressed such common sense issues. Additionally, changing eating behaviour is a huge societal challenge, it is difficult, deserves full attention and it should be addressed in a comprehensive manner. This paper simply shows that increasing availability of fruits and vegetables does not affect body weight, which is somehow expected, but it fails to provide a more in depth consideration of the many other factors that affect food choice. 

The paper is nice to read, very good specialist material, high level statistics, but lacks comprehensiveness.

Reviewer #3: This is a well-conducted study on the impact of a nationwide school fruit and vegetable policy on BMI or overweight of school children in Norway. The study design, datasets, and statistical methods and analyses are mostly adequate. The adjustement for confunding factors is key for the design of a quasi-natural experimental study and the authors addressed this well by adjusting adequately for region, population density, cohort, and parental education. However, there are still a few major issues needing attention especially on presentation of the results.

1) Overall the presentation of the results is a bit all over the place with too much technical details which became difficult to focus and follow. Many tables and figures could be put into supplementary information and only need briefly summarised in the main text. For example, table 1 is a huge table with too many details which is not very informative. Can either remove table 1 to supplementary or make it concise with key messages. Table 2 on pre-intervention can be remove to supplementary information as it's not key outcome. The same for Figure 3. It's only on a subgroup analysis but huge with many plots so can be removed to supplementary info.

2) Table 1. There are 4 cohorts but why pooled results only with 3 cohorts? Throughout the paper, especially in Figure 2, there are different combination of 3 of 4 cohorts analysed and pooled, which is a bit difficult to follow. Authors need to make the analyses consistent and neat. If not, please explain why and also what's the impact of using different cohorts on the analyses? Any bias or limitation?

3) The section of Population distributions and also Figure 6 can be removed to supplementary info as not key at all.

4) The paper could be improved to be more focused and concise with key messages presented. At the moment, the readers are overwhelmed with huge and complex tables and figures and technical details, many of which could be better placed in supplementary information. At the moment, there are so many outcomes and analyses presented but could authors consider if they can be classified/identified as primary or secondary outcomes so that the paper is focused on the key messages.

5) Figure 2 is a key figure. As the different cohorts have different time exposoure to FFV policy, have the pooled results been adjusted for this exposure difference in the analyses?

6) Missing data and children transferred between types of schools. Need a section specifically on the above issues on the details of missingness and ways to deal with the missingness and discussion of potential biases. It seems missing data were simply removed from the analyses but do we know the pattern of the missingness and what is the potential bias by excluding this missing data?

[LINK]

---

## [Decision Letter · Decision Letter 2]

25 Nov 2021

Dear Dr. Øvrebø,

Thank you very much for re-submitting your manuscript "A nationwide school fruit and vegetable policy and childhood and adolescent overweight: A quasi-natural experimental study" (PMEDICINE-D-21-04001R2) for review by PLOS Medicine.

I have discussed the paper with my colleagues and the academic editor and it was also seen again by two reviewers. I am pleased to say that provided the remaining editorial and production issues are dealt with we are planning to accept the paper for publication in the journal.

[LINK]

We look forward to receiving the revised manuscript by Dec 02 2021 11:59PM.   

Sincerely,

Callam Davidson, 

Associate Editor 

PLOS Medicine

plosmedicine.org

Requests from Editors:

Please check and update the URLs in your Data Availability Statement, as both lead to pages that no longer exist.

In your abstract, please quantify your main findings with 95% CI and p values (where appropriate).

Line 52: Please update to ‘The main limitations of this study are the potential for residual confounding and exposure misclassification, despite efforts to minimise their impact on conclusions’.

Line 67: This bullet can be deleted.

Line 69: Please update to ‘To promote a healthy diet, from 2007 to 2014 a nationwide free fruit and vegetable policy ensured a daily piece of free fruit or vegetable was available to all children in Norwegian combined schools (covering grades 1-10)’.

Please also consider including the age group covered by Norwegian combined schools, as grade age boundaries will differ between countries. 

Line 71: Please update to ‘Studies on the potential benefits or consequences of such fruit and vegetable policies are important in improving public health efforts to tackle childhood overweight and obesity’.

Line 81: Delete the sentence beginning ‘Data are from’ and instead relocate this to the start of the following bullet such that it reads ‘Using data from 11215 Norwegian children and early adolescents, we observed little…’

Line 143: ‘We did not register’.

Line 196: Please update to ‘Other data’

In Figures 3 and 5, please use the legend to denote which statistical test was used to derive the p values.

Please also check supplementary figures in light of the comment above.

Line 578: Update ‘little’ to ‘few’.

Line 661: The ‘Ethics approval’ section can be removed as it is now redundant. 

References 3 and 5: Please include (date accessed: DD/MM/YYYY).

Reference 41 contains unnecessary COI information, please remove this.

Please ensure all journal abbreviations in the references are consistent with those found in the National Center for Biotechnology Information (NCBI) databases.

Please update your STROBE checklist to use section and paragraph numbers as opposed to page numbers (which are liable to change during the publication process).

Comments from Reviewers:

Reviewer #1: The authors adequately addressed my concerns. I agree with the authors that post-hoc power analysis would not be informative. Thank you,

Stephan J Guyenet

Reviewer #3: Thanks authors for their great effort to improve the manuscript. I am satisfied with the response and revision. No further issues needing attention.

[LINK]

---

## [Editor Report · Decision Letter 3]

1 Dec 2021

Dear Dr Øvrebø, 

On behalf of my colleagues and the Academic Editor, Dr Barry Popkin, I am pleased to inform you that we have agreed to publish your manuscript "A nationwide school fruit and vegetable policy and childhood and adolescent overweight: A quasi-natural experimental study" (PMEDICINE-D-21-04001R3) in PLOS Medicine.

When making the formatting changes, please also make the following update:

* Line 91: Please correct 'polices' to 'policies'.

PUBLICATION SCHEDULE

Given our busy publication schedule for the remainder of 2021, we are planning to publish your paper in early 2022 (the exact date will be communicated to you once confirmed).

PRESS

Sincerely, 

Callam Davidson 

Associate Editor 

PLOS Medicine